# Pathomorphogenesis of Glycogen-Ground Glass Hepatocytic Inclusions (Polyglucosan Bodies) in Children after Liver Transplantation

**DOI:** 10.3390/ijms23179996

**Published:** 2022-09-02

**Authors:** Francesco Callea, Paola Francalanci, Chiara Grimaldi, Francesca Diomedi Camassei, Rita Devito, Fabio Facchetti, Rita Alaggio, Emanuele Bellacchio

**Affiliations:** 1Department of Pathology, Catholic University Bugando Medical Centre, Mwanza P.O. Box 1464, Tanzania; 2Pathology Unit, Department of Laboratories, Bambino Gesù Children’s Hospital, IRCCS, 00165 Rome, Italy; 3Department of Surgery, Children Hospital Meyer, 50139 Florence, Italy; 4Department of Pathology University of Brescia Spedali Civili, 25121 Brescia, Italy; 5Area di Ricerca Genetica e Malattie Rare, Bambino Gesù Children’s Hospital, IRCCS, 00165 Rome, Italy

**Keywords:** glycogen-ground glass, polyglucosan bodies, hepatocytes, drugs

## Abstract

Seventeen out of 764 liver biopsies from transplanted (Tx) livers in children showed glycogen-ground glass (GGG) hepatocytic inclusions. The inclusions were not present in pre-Tx or in the explanted or donor’s liver. Under the electron microscope (EM), the stored material within the cytosol appeared as non-membrane-bound aggregates of electron-lucent globoid or fibrillar granules, previously described as abnormally structured glycogen and identified as Polyglucosan bodies (PB). The appearance of GGG in our children was analogous to that of PB-GGG occurring in a number of congenital diseases due to gene mutations such as Lafora’s d., Andersen’s d., Adult Polyglucosan Body Disease and glycogenin deficiency. The same type of GGG was previously reported in the liver of patients undergoing transplants, immunosuppressive or antiblastic treatment. To explore the potential mechanism of GGG formation, we examined whether the drugs after whose treatment this phenomenon was observed could have a role. By carrying out molecular docking, we found that such drugs somehow present a high binding affinity for the active region of glycogenin, implicating that they can inactivate the protein, thus preventing its interaction with glycogen synthase (GS), as well as the maturation of the nascent glycogen towards gamma, beta or alfa glycogen granules. We could also demonstrate that PG inclusions consist of a complex of PAS positive material (glycogen) and glycogen-associated proteins, i.e., glicogenin-1 and -2 and ubiquitin. These features appear to be analogous to congenital GGG, suggesting that, in both cases, they result from the simultaneous dysregulation of glycogen synthesis and degradation. Drug-induced GGG appear to be toxic to the cell, despite their reversibility.

## 1. Introduction

Glycogen-ground glass (GGG) inclusions in the liver, as well as heart, brain and neuromuscular cells, consist of polyglucosan accumulation. They are observed in a group of genetically determined disorders classified as Secondary Glycogen Storage Disease (SGSD), as opposed to Primary GSD. The latter include: disorders of glycogen synthesis, disorders of glycogen breakdown (glycogenolysis), disorders of glycolysis and lysosomal glycogenosis [1].

Secondary GSD comprise enzyme deficiencies caused by gene mutations such as Lafora’s disease, Andersen’s disease, Adult Polyglucosan Body Diseases (APBD), glycogenin deficiency and a variety of myopathies [1,2,3].

The secondary GSD are characterized by storage of polyglucosans (macromolecules or polymers of non-branched insoluble glycogen) with a peculiar EM appearance, consisting of electron-lucent globular or fibrillar material. The stored material is composed of a complex of non-branched glycogen and glycogen-associated proteins (glycogen proteome) [4].

Glycogen synthesis involves a main enzymatic pathway beginning with the intramolecular transfer of the glucose bound to uridine-5′-diphosphate (UDP) present on one glycogenin, the initiator of glycogenesis [5] to the Tyr194 residue in another glycogenin, and subsequent sugar chain intramolecular elongation. Thereafter, glycogenin interacts with glycogen synthase (GS) for further glucose monomer elongation, and in turn, GS interacts with the glycogen-branching enzyme (GBE) for branching and forms the mature glycogen molecule [5].

The discovery of glycogenin as the initiator of glycogenesis and its capacity for self-glycosylation [6] and of proteoglycogen [7] have contributed to understanding the maturation process of glycogen gamma particles into beta and alpha granules [8], as well as the arrest of maturation and storage in the form of PB (Figure 1).

Normal glycogen maturation is hampered by the arrest of interaction or mutation of glycogenin and glycogen synthase (GS) or glycogen-branching enzyme (GBE). Alpha or beta granules (mature glycogen) are not formed. Proteoglycogen and glycogenin accumulate as PG.

Self-glycosylation of glycogenin-1 can trigger glycogenesis with no need for the main pathway. Another self-glycosylation protein, glycogenin-2, is involved in liver glycogen biosynthesis [9].

As a general rule, dysregulation of the anabolic or catabolic pathway leads to storage of glucose as glycogen [6].

Dysregulation of glycogen-associated proteins causes storage of polyglucosans. Glycogenin deficiency [3], as well as mutations of GS [10] or GBE [11] result in PG storage in the form of GGG (Figure 1). The same holds true for Lafora’s and Andersen’s diseases [1].

GGG hepatocytes were first reported in an adult patient after renal transplantation [12]. The GGG hepatocytes showed the same light and EM appearance as the genetically determined GGG [12]. Since 2006, an increasing amount of GGG has been reported in paediatric Tx livers, as well as in patients undergoing immunosuppressive, antiblastic or antiviral therapy or in patients with underlying liver disease [13,14,15]. These observations have raised the suspicion that the GGG phenomenon in these categories of patients could be related to polypharmacy [13,14,15,16].

However, GGG has also been reported in children under prolonged parenteral nutrition with or without immunosuppressive treatment [13,14,17].

Whatever the aetiology, it appears that the formation of GGG takes place through a disturbance of the glycogen metabolism.

Due to the striking morphological similarity of acquired and congenital GGG, our investigation on the 17 cases developing GGG was addressed to the pathway leading to the formation of PG bodies. A molecular docking analysis was carried out to evaluate whether the drugs, after which GGG had developed, could form stable complexes with glycogenin, thus altering its function in analogy to enzyme deficiency or gene mutation conditions.

In this paper, we discuss the morphological and immunohistochemical findings in light of the molecular docking results.

## 2. Results

Routine histology:Group 1 and 2.No specimens from these groups were found to contain GGG.Group 3.

Seventeen liver biopsy specimens from twelve post-Tx (10 liver/2 HSC + liver) children contained GGG. Five patients underwent more than one biopsy during the follow up.

A single patient (patient n. 7 from Table 1 and Table 2) underwent five liver biopsies in seven months (Table 2). Three biopsies contained GGG; the first biopsy was obtained 6 months after Tx. GGG was also observed in two subsequent biopsies obtained within two months. The last two follow-up biopsies did not contain GGG (Table 2). 

Histological findings are shown in Panel A (Figure 2).

In all cases, including the three specimens from patient n.7 (Table 2), GGG hepatocytes were mainly concentrated in zone 1 (Figure 2a) and appeared in the form of eosinophilic GGG inclusions, filling up the entire cytoplasm, or part of it, in a crescent-like fashion (Figure 2b). All inclusions were PAS positive (Figure 2c × 40) and PASD negative (Figure 2d × 40). In serial immunostained sections, GGG inclusions showed positivity for glycogenin-1 (Figure 2e × 40), glycogenin-2 (Figure 2f × 40) and ubiquitin (Figure 2g × 60). In biopsy n.3 from the same patient, ubiquitin positivity was restricted to cytoplasm and subplasmalemmal membrane; nuclei were negative (Figure 2h × 100).

Control sections (omission of the primary antibodies and pre-absorption of antibodies with purified antigens) were negative. 

Electron microscopy findings (Figure 3 Panel B).

Under the EM, GGG cells contained plenty of globoid, minute or coarse, pale (electron-lucent) material without recognizable trend to alpha or beta granule formation (Figure 3a).

In biopsy n.2, the electron-lucent material was occupying half of the cell cytoplasm and displaced the organelles towards the remaining half (Figure 3b). This phenomenon was better appreciated in adjacent hepatocytes (Figure 3c). In the third biopsy, individual granules acquired a star-like contour, mimicking non-confluent alpha or beta granules (Figure 3d). Several hepatocytes showed signs of cytoplasmic damage: increase in the number and polymorphism of lysosomes (Figure 3b), mitochondriophagy (Figure 3c), autophagic vacuoles (Figure 3c,d) and megamitochondria (Figure 3e). Moreover, apoptotic bodies were seen in the sinusoids or within the cytoplasm of hypertrophic Kuppfer cells (Figure 3f). Apoptotic bodies contained compact aggregates of dense unstructured granules.

Three-dimensional Modelling studies.

The best docking poses of the examintwed drugs (tacrolimus, bactrim, deursil, MFM, aspirin and acyclovir) on apo-glycogenin-1 structure are shown in Figure 4. The calculated binding affinities (reported as the negative logarithms of the dissociation constants) for these complexes were as follows, in decreasing order: tacrolimus, 9.25 (highest affinity); bactrim, 8.42; deursil, 8.22; MMF, 6 61; aspirin, 5.58; acyclovir, 4.27 (lowest affinity).

## 3. Discussion

The results from this study can be summarized as follows:1.GGG have been demonstrated in 17 paediatric liver biopsies at different intervals after liver Tx.2.GGG consisted of a complex of glycogen-associated proteins and polysaccharides less branched than in normal glycogen.

The polysaccharides accumulate in the form of PG bodies. Hepatocytes containing PG bodies show evidence of cell damage.3.Three kinds of proteins have been detected within post-Tx-acquired GGG: glycogenin-1, glycogenin-2 and ubiquitin.4.Post-Tx GGG inclusions appear to be drug-induced and reversible.5.Drug induction can be explained as an effect of an undue occupation of the region of glycogenin adhibited to the binding of UDP and sugars.6.Light microscopy and immunohistochemical features of acquired GGG were quite analogous to the genetically determined GGG.7.Heaps of non-maturing native glycogen (proteoglycogen) appear as coarse electron-lucent globular or filamentous material.8.The same protein content and EM appearance of the stored material in congenital and acquired GGG, indicates that the storage process could follow the same pathway.9.The arrest of the sequential glycogenin–GS–GBE interactions leads to insoluble PG.

The metabolic implications of PG storage appear to be related to the unique property of the glycogen-associated proteins that not only interact, but also work in tandem in this biological process [1,4,18]. Inactivation or mutations of any glycogen-associated protein (glycogenin, laforin/malin, GS, GBE, RBCK1, the latter causing Polyglucosan Body Myopathy-1), provoke simultaneous accumulation of glycogenin and ubiquitin. The observation that dysregulation/mutation of GS or GBE is accompanied by accumulation of glycogenin points out to the crucial role of glycogenin in initiating both the normal glycogenesis and the abnormal diversion towards the proteoglycogen and PB as shown in Figure 1. At the same time, it appears that upstream alterations are reflected backwards.

Equally, in Lafora’s disease, the prototype of PG storage disease, the dysregulation/mutations of the laforin/malin complex provoke the combination of GS dysregulation and impaired autophagy [1,18]. It is noteworthy that we have found the combined accumulation of glycogenin and ubiquitin in the liver of our Tx children in perfect analogy with glycogenin deficiency [3].

In summary, the anabolic implication of PG storage is easily understandable, as the interruption of the sequential glycogenin–GS–GBE chain of interactions blocks the maturation of glucose monomers/polymers to the stage of native glycogen, which represents proteoglycogen and, to some extent, possibly gamma particles.

The catabolic implications depend at least upon two reasons: (i) the insolubility of the PG polymers, and (ii) the inactivation of the ubiquitin–proteasome and autophagy systems.

When comparing the results from our study and the GGG phenomenon in Secondary GSD, we find two main potential differences.

First, in our GGG, ubiquitin can show nuclear staining that is not described in secondary GSD. 

The positive nuclear staining is intriguing but could be explained by the property of the nuclear ubiquitin–proteasome system (UPS): visualization of proteasome, protein aggregates and proteolysis in the cell nucleus [19]. The nuclear UPS plays, among others, a role in multiple cellular events, including signal transduction and transcription [19,20,21].

Secondly, in secondary GSD, the PG storage is systemic and can affect the muscle, heart, nerves, brain, endocrine glands, as well as the liver [1,2,3,11].

As far as we know, no systematic studies have been performed in extrahepatic sites in Tx patients with hepatocytic GGG. In a single case, investigations were performed to exclude a multisystem disease [17].

In our docking study on the possible complexation of glycogenin by drugs, the calculated binding affinity (reported as the negative logarithms of the dissociation constants) for the top-scoring molecular poses were as follows (in decreasing order): tacrolimus, 9.25 (highest affinity); bactrim, 8.42; deursil, 8.22; MMF, 6 61; aspirin, 5.58; acyclovir, 4.27 (lowest affinity). Notwithstanding the difficulty in inferring accurate protein–ligand complex structures by docking, these results support the idea of drug interactions with glycogenin. In fact, the relatively high predicted affinities suggest that some of the drugs could bind to this protein upon achievement of sufficient concentrations. Potential interactions with drugs pose a high risk, particularly to the liver, because due to the detoxifying role of this organ, it acts as a concentrator of exogenous substances, thus shifting chemical equilibria towards the formation of complexes between drugs and host proteins.

Undue interactions of drugs with the glycogenin region that normally host its natural ligands can impair the function of this protein. Consequentially, drug-induced glycogen dysregulation can manifest pathological features similar to those observed in enzyme deficiency or gene mutations, albeit reversibly upon cessation of pharmacological treatments.

Although these results must be considered preliminary, they provide plausibility that at least tacrolimus, bactrim, deursil and MFM can bind with significant strength to glycogenin, thus interfering with its function. That of course, according to the knowledge of the glycogen metabolism, can provide a possible explanation for the accumulation of sugars in transplanted children treated with these drugs. Among the latter, tacrolimus appears to be the most powerful.

To try to understand whether GGG inclusions are a sign of hepatocytic injury and not of adaptation, we have compared our results with GGG inclusions regularly formed in the liver of Antabuse [22] or Cyanamide [23] treated alcoholic individuals, as well as in experimental animal models [24,25].

Similar to the observation in experimental animal models, hepatocytes with PG bodies in our Tx patients have shown an increased number and polymorphism of lysosomes and disruption of endoplasmic reticulum (ER) membranes (Figure 3d), autophagic vacuoles (Figure 3c,d), mitochondriophagy (Figure 3c), megamitochondria (Figure 3e) and apoptotic bodies (Figure 3f).

In contrast, hepatocytes without GGG inclusions showed structured alpha and beta glycogen granules in the absence of cytosolic or organelle damage. The GGG features are interpreted as an expression of cell injury, i.e., elementary lesions as the morphological counterpart of the disturbed glycogen metabolism. 

At the present time, however, the clinical significance of these alterations in terms of potential capability of causing transient serum transaminase elevation is unclear.

This, however, is not sufficient to suggest variations in the treatment, as the main purpose of clinical transplants is to save the graft rather than preventing transient, non-progressive histological changes. Likewise, while it seems reasonable to try to alleviate the irreversible permanent PG storage in secondary glycogen storage disease (SGSD) by using molecules that bind to the lysosomal membrane protein (LAMP-1) [26] in order to enhance autophagic glycogen catabolism [27], this would not be recommended for a non-progressive and reversible lesion such as drug-induced GGG.

With regard to the outcome of GGG in our series, we could analyse a follow-up biopsy in a single patient, as we observed persistence of GGG over two months and then disappearance. In the literature, there are very few cases with more than one biopsy; the longest duration is 36 months [14]. Interestingly, in one child, there was a 40-month gap between the suspension of immunosuppressive drugs and the reinstitution of parenteral nutrition (PN) [17]. This would suggest that PN alone is a significant factor [17]. In our series, four patients had been on both PN and immunosuppressive drugs.

Obviously, to unravel the lifespan of GGG, it would be necessary to examine serial biopsies, which is only possible in experimental animal models [24].

Likewise, it is not possible to calculate the incidence/prevalence of the GGG phenomenon outside a given series. In our study, the phenomenon has occurred in 17 out of 764 liver biopsies. All other studies report individual or group cases [14,15,16,17]. In conclusion, this study has proven that the GGG phenomenon reflects a disturbance of the glycogen metabolism and that the pathway recapitulates that of the PG body formation in SGSD.

Further studies are needed to clarify: (i) the higher prevalence of the GGG phenomenon in children compared to adults; (ii) the mechanism of GGG formation in non-drug-induced GGG, especially in children on parenteral nutrition for which the link could be conceivable; (iii) whether GGG changes represent a true elementary lesion and, as such, a sign of cell injury; (iv) whether PG storage can occur outside the liver in drug-induced GGG and (v) why GGG occurs mainly or exclusively in zone 1 hepatocytes.

## 4. Material and Methods

The material of the study comprised three groups of patients:Group 1: 764 liver biopsy specimens obtained for diagnostic purposes from 267 liver-Tx performed at Bambino Gesù Children’s Hospital, Rome, in the period 2008–2021.Group 2: 12 explanted and 12 donor livers.Group 3: 17 needle liver biopsies from 12 children who had displayed GGG after liver Tx.

Clinical data from Group 3 are given in Table 1 and Table 2.

All specimens were formalin fixed and paraffin embedded. Four-micron tick histological sections were coloured with H&E, PAS and PAS after amylase digestion (PAS-D).

From GGG-containing specimens, serial sections were cut and stained via standard immunohistochemistry for glycogenin-1 (Atlas Antibodies AB, SE-223 63 Lund, Sweden, dilution 1:100) and glycogenin-2 (same source, dilution 1:200). Immunostaining for ubiquitin was performed using the anti-ubiquitin rabbit polyclonal antibody (Agilent technologies, Milan, Italy), diluted 1:300 in PBS buffer, without antigen retrieval. After 60 min of incubation, the Novolink Polymer (Leica, Biosystem, Milan, Italy) was applied for 15 min and the immunoreaction was revealed using diaminobenzidine as chromogen. Sections were counterstained with Meyer’s haematoxylin.

Semithin sections were cut from Karnowski fixed Epon or dewaxed material and stained with methylene blue to localize the GGG. An ultrathin copper-mounted grid was examined under a Zeiss EM.

Molecular complexes of the selected drugs with glycogenin were obtained via docking with PLANTS [28], employing flexible ligands and protein side chains. As a protein structural target, the apo-form of the homodimerized glycogenin-1 (crystal structure with Protein Data Bank, PDB, entry 3Q4S) was chosen. Missing loops in the crystal structure (disordered amino acids, a.a. 194–196 and 234–242) were modelled.

We carried out the docking search within 30 Å of the centre of the protein empty space determined with Hollow [29], which also contains the binding pockets of the natural ligands of glycogenin, uridine-5′-diphosphate (UDP) and sugars (Figure 5).

Binding affinities for the docked drug–glycogenin complexes were calculated with Cyscore [30].

The drugs examined in the docking were those after whose treatment patients had developed the GGG hepatocytic inclusions: tacrolimus, bactrim, deursil, mycofenolate mofetil (MMF), aspirin and acyclovir (Table 1).

## Figures and Tables

**Figure 1 ijms-23-09996-f001:**
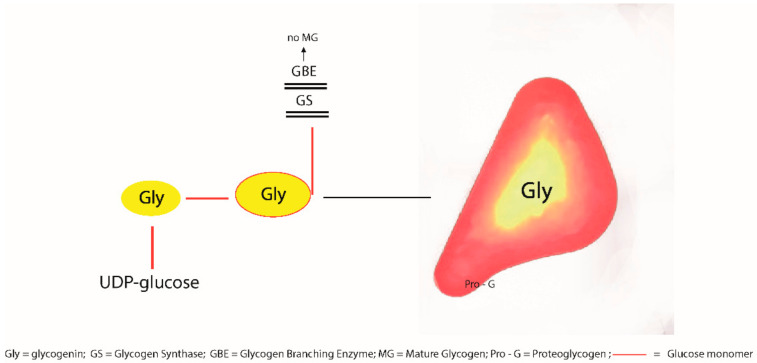
Schematic representation of the destiny of glycogenin after glycosylation (self- or enzymatic glycosylation).

**Figure 2 ijms-23-09996-f002:**
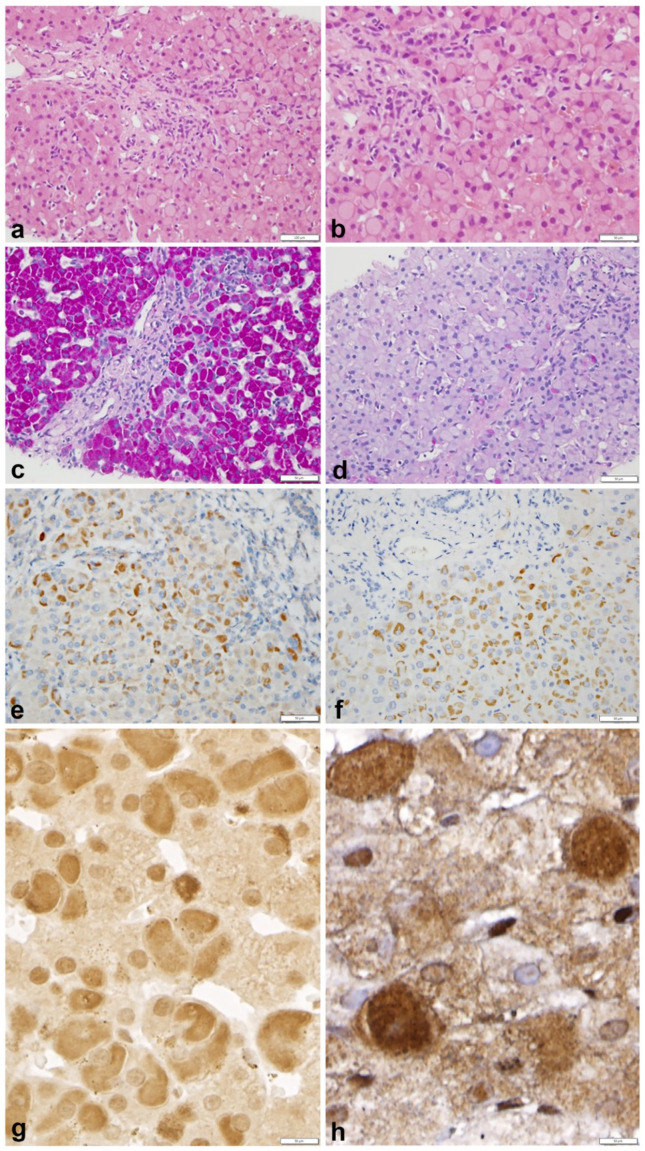
Panel A. Serial sections of liver biopsy n.2 from patient n.7. GGG are mainly located in zone 1 and appear as pale, eosinophilic (**a** H.E. × 20) inclusions occupying the entire cytoplasm, or part of it, in a crescent-like form (**b** HE × 40). The inclusions are strongly PAS positive (**c** PAS × 40) and negative on PAS after diastase (**d** PAS-D × 40). GGG inclusions are positive on immunostaining for glycogenin-1 (**e** × 40) and for glycogenin-2 (**f** × 40). Ubiquitin stain shows full cytoplasmic or crescent-like positivity in all inclusions and in a few nuclei (**g** × 60). In biopsy n.3 and 4 from the same patient, the ubiquitin shows only cytoplasmic and subplasmalemmal membrane positivity. Nuclei are not stained (**h** × 40).

**Figure 3 ijms-23-09996-f003:**
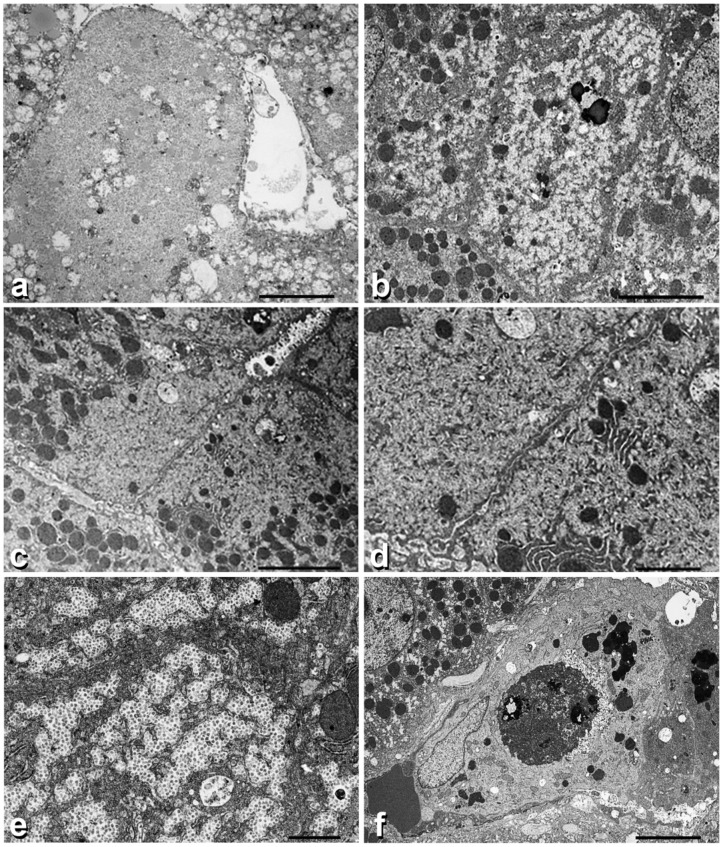
Panel B. Electron micrographic pictures of biopsy n.2 and 3 from the same patient showing GGG. (**a**) shows a hepatocytic cytoplasm completely filled up by packed coarse electron lucent material. (**b**) shows similar material in half of the cytoplasm, corresponding to the crescent-like appearance in light microscopy. Organelles are marginated towards the opposite half. This pattern is better appreciated when observing two adjacent hepatocytes (**c**). (**d**) shows two adjacent hepatocytes containing auyophagosomes at the apical poles. (**e**) shows the cytoplasm of a hepatocyte containing star-like material resembling alpha or beta granules that compress the endoplasmic reticulum membranes. An autophagic vacuole and a megamitochondrion is also seen. (**f**) shows a hypertrophic Kuppfer cell containing several lysosomes and an apoptotic body. The apoptotic body shows a dark appearance (electron dense) of all organelles as well as of the compact aggregates of amorphous, somewhat globular material, corresponding to PG.

**Figure 4 ijms-23-09996-f004:**
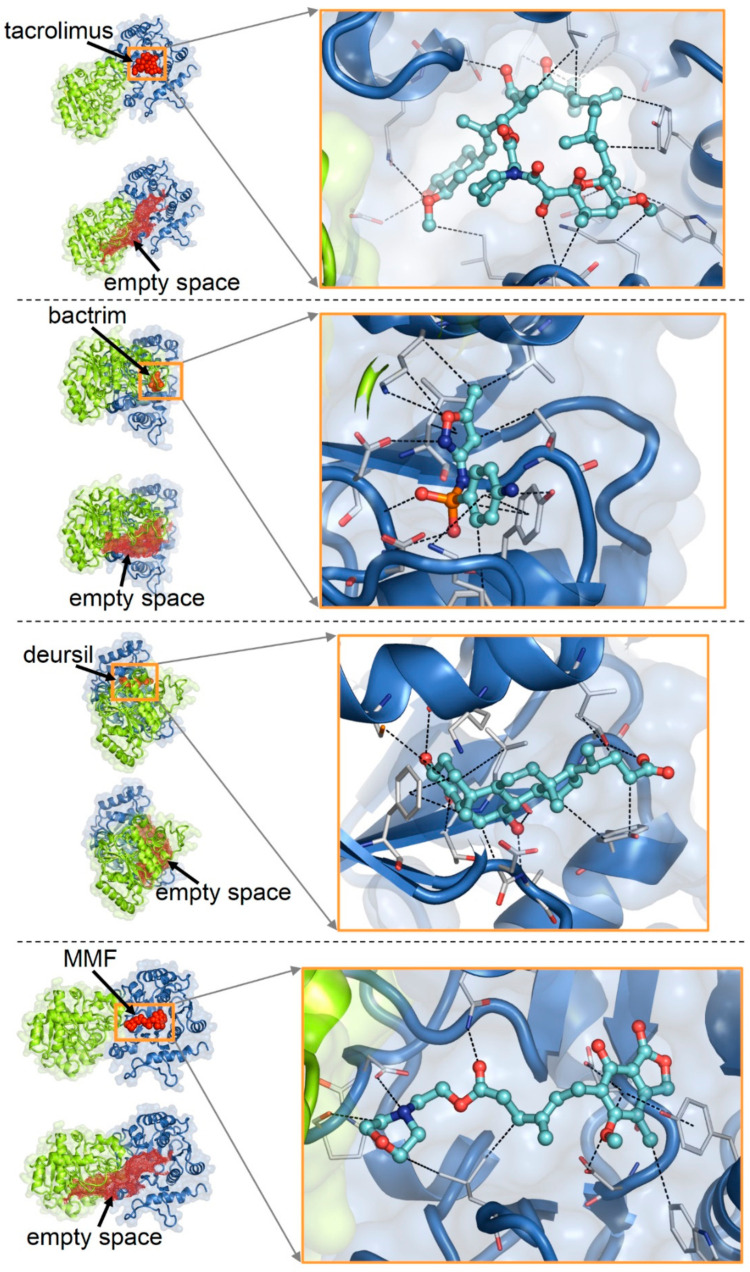
Docking poses of tacrolimus, bactrim, deursil, MMF on the structure of the homodimeric apo-glycogenin-1. For each drug, the whole protein–drug complex, the protein with the calculated empty spaces, a zoomed view of the bound drug (balls and sticks) and its interactions (dotted lines) with amino acids (thin sticks) are shown.

**Figure 5 ijms-23-09996-f005:**
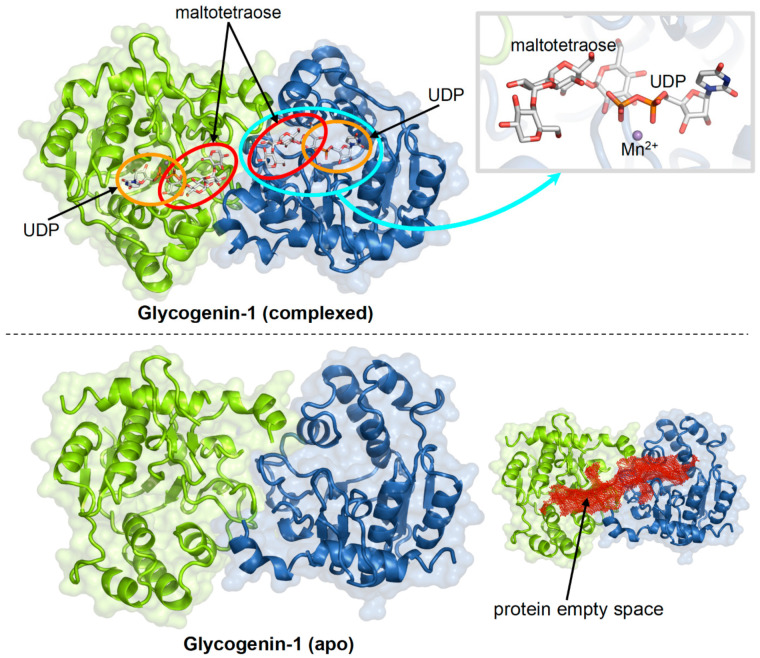
Crystal structures of the homodimeric human glycogenin-1 complexed with UDP and maltotetraose (PDB 3U2U; the binding regions of ligands are highlighted by ovals) and the apo protein (PDB 3Q4S; the interior empty space of the protein is indicated by red dots).

**Table 1 ijms-23-09996-t001:** Clinical data from 12 patients with GGG in liver biopsies.

Case	Age/Sex	Pathology	Interval LTx-GGG	Rejection & Treatment	Recent Medications	ALT/AST/GGT	New Biopsy & GGG	PN
1	9y/F	WD	3m	Y Steroid boli	TAC, deursil, aspirin, antibiotics	90/36/16	N	N
2	1y/F	BA	11m	N multiorgan failure	TAC, deursil, anti-epilectics	normal	N	Y
3	1y/M	BA	6m	Y Steroid boli	TAC, deursil, aspirin, antibiotics	normal	No biopsy	N
4	8y/M	HyperIgG + cryptosporidium	2m	N HSCT-GVHD	Multidrug treat	488/266/780	N	Y
5	9m/F	BA	3y 10m	Y Steroid boli	Bactrim, aciclovir	101/71/20	No biopsy	N
6	7m/F	BA	23m	Y Steroid boli	TAC, MMF, deursil	44/36/63	No biopsy	N
7	15m/M	Primary hyperoxaluria	6m	N Steatosis/viral acute hepatitis (HHV6)	Prograf, deursil	144/141/73 731/1122/142 103/93/101	Y in 2 other biopsies	N
8	2y8m/M	Leucinosi	2y	N Steatosis (incorrect diet)	TAC, deursil, anti-epilectics	33/50/20	No biopsy	N
9	8m/F	BA	6m	N	TAC, deursil, aspirin, antibiotics	normal	No biopsy	N
10	6y/M	BA	1y	Y Steroid boli	TAC, MMF, deursil	normal	N	N
11	4y/M	BA	1y,7m	N	TAC, deursil	167/96/13	N	N
12	4y/M	Refractory cytopenia	10m	N GVHD in HSCT	Multidrug treatment	60/136/835	N	N

WD = Wilson disease; BA = biliary atresia; Y= yes; N = no; TAC = tacrolimus; MMF = micofenolate mofetil; ALT and AST = transaminases; GGT = gammaglutamyltransferase; HSCT = haematopoietic stem-cell transplantation; GVHD = host versus graft disease; TMO = bone marrow transplantation; HHV6 = hepatitis herpes virus 6; PN = parenteral nutrition; Y = yes; N = no.

**Table 2 ijms-23-09996-t002:** Patients with GGG undergoing follow-up biopsies. A single patient (n. 7 from Table 1) presented GGG in three biopsies.

N Tx	N Biopsy	GGG	Interval Tx-GGG	Repeated Biopsies	Persistent GGG
217	764	15	3 mo–3 yrs (5 < 6 mo)	4	1 (patient N. 7)
Patient N. 7					
Age	Disease		Interval Tx-GGG	Date of biopsy	GGG
15 mo	Primary hyperoxaluria		6 mo	24 April 2018	++
				30 April 2018	+++
				14 June 2018	+
				28 November 2018	-
				9 January 2019	-

## Data Availability

Childrens’ Hospital Bambino Gesù, Rome, Italy.

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
