# Peer review of "Pathomorphogenesis of Glycogen-Ground Glass Hepatocytic Inclusions (Polyglucosan Bodies) in Children after Liver Transplantation"

_ijms, 2022, doi:10.3390/ijms23179996_

Round 1

Reviewer 1 Report

The article entitled “Pathomorphogenesis Of Glycogen-Ground-Glass Hepatocytic Inclusions (Polyglucosan Bodies) In Children After Liver Transplantation” is not a unique observation in these types of patients. Glycogen-Ground-Glass (GGG) Hepatocytic Inclusions have been detected in other pathological conditions including liver diseases. However, the study has been accomplished in a systemic manner and that provides some scientific clues regarding these GGG inclusions. The following point should be clarified:

1.      The long-term outcome should be discussed in detail. Do these bodies persist for a prolonged period or not?

2.      The authors claimed that these are toxic. Please provide the logic and evidence for these types of expressions.

3.      These bodies have been found in children. What about prevalence in immune-suppressed individuals?

Author Response

We are grateful to the Reviewer especially for his request to improve the items under score (X):

(Introduction = line 79-94; discussion metabolic pathways = 234-255). The number of references has increased by five.

With regard to the general comments, we have used the collected series and tried to clarify the possible mechanism of formation of GGG (line = 234-262). To do that we have compared our findings with those of GGG occurring in Secondary Glycogen Storage Disease (line = 244-248)and with those observed in experimental animal models (line = 300-314)       and evaluated by  molecular docking the binding affinity of the tested drugs with the active regions of glycogenin. This methodology is appropriate to study  drug-mediated effect.

We believe that the results of the study will be helpful to investigate the mechanisms of GGG formation in conditions other than transplantation.

With regard to the specific points n.1,2 and 3, the answer to the questions have been given in the revised text marked as track changes (n. 1 long term outcome = line 325-334, n.2 toxicity  = line 300-314; prevalence in immunosuppressed patients = 334-337)

Reviewer 2 Report

Dear Authors :),

Your paper is written using correct English, the subject is interesting & quite innovative, but in this form rather for lab scientists than liver transplant clinicians - and here is my main question / issue - Do you know about any clinical importance of your discovery for the patients ? 

I'm asking about it, because you write about elevated AT enzymes in patients GGG(+), but there is no evidence / analysis of this description in your article (lines 272-274) ? If it was more significant in GGG(+) group than in GGG(-) group ??? If you could calculate this, maybe it would be very good clinical conclusion for "the higher level" paper ??? :)

Minor problems are:

- no abbreviations description to the table no. 1

- line 140 - missing "G" letter :)

- line 212 - double "and" :)

- line 229 - no abbreviation description for "PBCK1" ... I haven't found ???

Best Wishes :)

Author Response

We wish to thank the Reviewer for his appretiation of our work and for remarking the innovative aspects.

The scope of the study and the research design are more clearly presented (line = 79-94) in the revised manuscript   Also the excursus on metabolic pathways of GGG formation is simplified  (line = 234-255)We believe that the applied methodology will turn out to be useful to study the GGG formation in conditions other than transplantation.

The number of references has increased by five.

We hope that our comment on the question about the clinical importance will be satisfactory (track changes, line = 315-324 ).

Indeed our paper was not meant to have a clinical scope but rather a scientific impact.

In the vrevised manuscript we have reported detailed ultrastructural alterations and classified them as elementary lesions. As such, they could be a sign of cell injury and the morphological counterpart of metabolic alterations.

Given that, we have used prudence in correlating the transaminase elevation with the morphological alterations (line = 300-314), as well as in attempting to calculate the difference among the GGG+ and GGG- cases.

Minor problems:

  • Abbreviations for Tab 1 : done
  • Line 140 : done
  • Line 212 : done
  • Line 229 : done

Round 2

Reviewer 1 Report

No specific comment.